# ImageNot: A contrast with ImageNet preserves model rankings

**Olawale Salaudeen**[*]                                                *olawale@mit.edu*
*Massachusetts Institute of Technology*

**Moritz Hardt**
*Max Planck Institute for Intelligent Systems, Tübingen*
*Tübingen AI Center*

**Reviewed on OpenReview:** *https://openreview.net/forum?id=YVbhMerXv9*

## Abstract

We introduce ImageNot, a dataset constructed explicitly to be drastically different than ImageNet while matching its scale. ImageNot is designed to test the external validity of deep learning progress on ImageNet. We show that key model architectures developed for ImageNet over the years rank identically to how they rank on ImageNet when trained from scratch and evaluated on ImageNot. Moreover, the relative improvements of each model over earlier models strongly correlate in both datasets. Our work demonstrates a surprising degree of external validity in the relative performance of image classification models when trained and evaluated on an entirely different dataset. This stands in contrast with absolute accuracy numbers that typically drop sharply even under small changes to a dataset.

## 1 Introduction

The Achilles heel of machine learning is its apparent lack of robustness. Models developed in one domain consistently perform worse in new domains. A slew of important—but essentially negative—empirical results demonstrated that even small changes to a dataset can impact the accuracy of machine learning models dramatically (Recht et al., 2019; Taori et al., 2020; Miller et al., 2021). Consequently, there is good reason to fear that machine learning lacks *external validity* (Torralba and Efros, 2011; Liao et al., 2021).

Against this pessimistic backdrop, we show that key qualities of machine learning models do generalize to a surprising degree. The starting point of our work is a thought experiment. How different a dataset from ImageNet would have supported essentially the same model developments over the years? ImageNet was at the center of the deep learning revolution in computer vision of the 2010s. Numerous deep architectures were developed specifically with ImageNet in mind. It's therefore reasonable to conjecture that a different benchmark would have spurred different model developments. Our investigation suggests otherwise.

To argue this point, we create a contrast with ImageNet, a new dataset called ImageNot. ImageNot has 1000 classes with 1000 examples per class, just like ImageNet's ILSVRC-2012 distribution, but ImageNot differs in key characteristics associated with ILSVRC-2012 (Russakovsky et al., 2015; Deng et al., 2009). No human annotators provided any labels. ImageNot pictures instead come from noisy web-crawled data selected based on the image captions and class names alone. While the classes in ILSVRC-2012 have strong concept overlap, the ones we chose for ImageNot are deliberately arbitrary and unrelated.

The central question we ask is: *Does ImageNot provide the same model rankings as ImageNet?* An important difference between our work and previous work is that they investigate the robustness of *ImageNet models*, models trained on ImageNet. In contrast, we investigate the external validity of the core deep learning

---

[*]Work done while interning at the Max Planck Institute for Intelligent Systems, Tübingen and as a PhD student at the University of Illinois at Urbana-Champaign.

developments on ImageNet; thus, we remove the ImageNet confound, unlike in previous work. The models developed on ImageNet are developed from scratch and evaluated on ImageNot. More generally, *we identify five major confounds that hinder interpretations of previous work in the same context of this work on external validity*: Previous work had (i) evaluation sets with same images as ImageNet but with corruptions (Hendrycks and Dietterich, 2019; Hendrycks et al., 2021; 2020), (ii) the same data sources, Flickr (Recht et al., 2019), (iii) the same image classes (Recht et al., 2019; Barbu et al., 2019; Hendrycks and Dietterich, 2019; Hendrycks et al., 2021; 2020), and (iv) the same data generative mechanisms, i.e., the procedure of human annotation (Recht et al., 2019). Furthermore, (v) the same models trained on ImageNet are evaluated (Recht et al., 2019; Barbu et al., 2019; Hendrycks and Dietterich, 2019; Hendrycks et al., 2021; 2020). While the results of these previous works have been interpreted in the context of the external validity of deep learning model development, i.e., scientific progress on ImageNet, these confounds limit the soundness of such interpretations, as they may account for the observed robustness of rankings. These confounds leave the question of the external validity of ImageNet as a benchmark for model development progress unanswered. Our study of external validity is scoped. We do not study whether ImageNet-era model rankings hold in the context of shifts like cross-domain, cross-task, detection, segmentation, long-tail, or open-vocabulary. Furthermore, throughout the paper, when we refer to historical ImageNet architectures or model developments, we mean this to broadly include the training procedures and norms by which they were historically introduced and evaluated. We enumerate the accompanying procedures in Appendix A. Our goal is to test whether the historical trajectory of ImageNet-era model development would have been similar on ImageNot, not to separate architecture from training recipe under a unified modern protocol.

The goal of this work is not to propose ImageNot as a new general benchmark or replacement for ImageNet. *ImageNot is a construct, removing these confounds, to address this question and stress test the external validity of model development progress on ImageNet.* Additionally, our work focuses on benchmarks as a tool for advancing science; there are many other interpretations of benchmarks that one may be interested in, e.g., as instruments for producing valid scores of a model's capabilities. Our demonstration of the robustness of rankings under certain benchmark properties, e.g., noise, should not be assumed to extend to other interpretations of benchmarks.

In a nutshell, we give a strong affirmative answer to this question of the external validity. Key model developments from the ImageNet era rank identically on ImageNot as they do on ImageNet. Moreover, each model's relative improvement compared with prior models is approximately the same on both datasets. Our empirical findings provide strong evidence for the external validity of model rankings.

## 1.1 Our contributions

Our primary contribution is an in-depth investigation into the external validity, robustness, and reproducibility of model development progress in machine learning. We create a new dataset and perform compute-intensive training and evaluation of numerous architectures on the new dataset.

**Creating ImageNot.** We created ImageNot from the LAION-5B dataset, a public collection of 5.8 billion image-text pairs crawled from the web and filtered by the CLIP model (Schuhmann et al., 2022; Radford et al., 2021). Although CLIP is trained with knowledge of ImageNet, the sheer size of LAION-5B ensures that images in LAION-5B generally are not similar to ImageNet images. We choose 1000 classes from the WordNet lexical database while avoiding nouns close to those in ImageNet. For each class, we pick 1000 images from LAION-5B by caption-class similarity alone. We use Roberta text embeddings for this purpose. As a text-only model, Roberta was never trained on ImageNet. This mitigates the concern that our selection of images from LAION-5B might be biased in favor of ImageNet-like instances. We give a detailed analysis of the differences between ImageNet and ImageNot. Crucially, our creation of ImageNot avoids confounds in previous work with respect to images and classes, data source, and generative mechanism. We provide detailed instructions for deriving ImageNot from LAION-5B in the provided code.

**ImageNot preserves model rankings and relative improvements.** Our primary empirical finding is that ImageNot has the same image classification model rankings as ImageNet. This finding is important because it provides evidence that the model architecture improvements centered around ImageNet could have

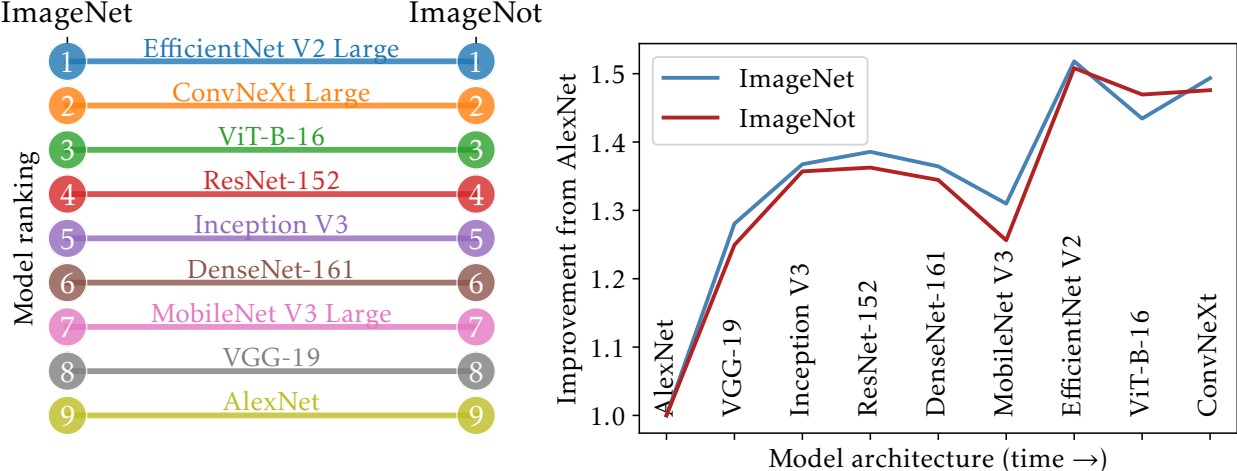

Figure 1: Final Model Ranking and Relative Improvement. Model rankings and relative improvements hold with models trained from scratch on the respective dataset, and evaluated on a held-out test set. For a set of test accuracies $X$, relative progress is $X_i/X_{\text{AlexNet}}$. ImageNet model accuracies range from 0.57 (AlexNet) to 0.85 (EfficientNet V2 L), while much noisier ImageNot model accuracies approximately range from 0.4 (AlexNet) to 0.6 (EfficientNet V2 L).

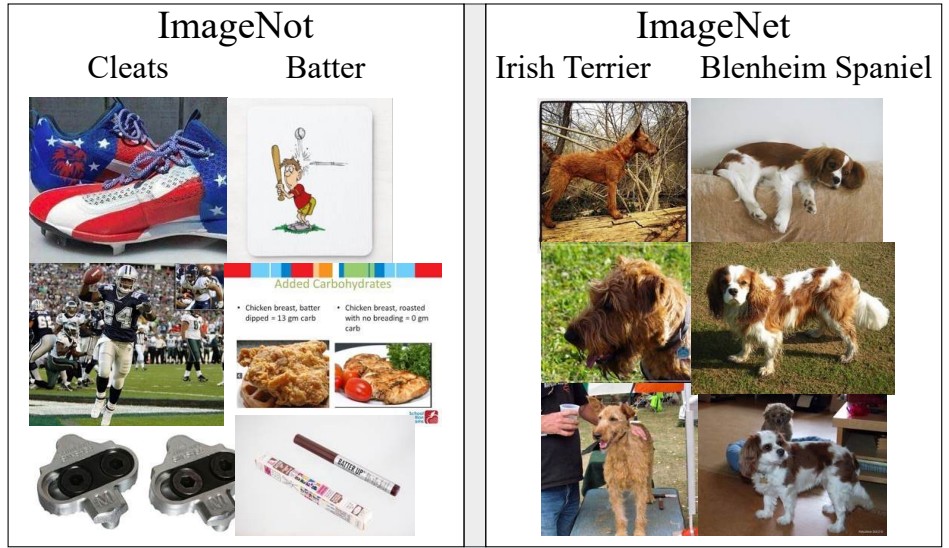

Figure 2: Examples from ImageNet (left) and ImageNot (right). ImageNot classes (e.g., 'cleats,' 'batter') are disjoint and distinct from ImageNet classes (e.g., 'Irish Terrier,' 'Blenheim Spaniel'). ImageNot, web-sourced data with automated data selection and labeling, also has greater natural variability and label noise.

occurred similarly for a different dataset than ImageNet. We consider 9 representative model architectures from AlexNet to Vision Transformers (Krizhevsky et al., 2012; Dosovitskiy et al., 2020). We train from scratch and evaluate each model on ImageNot. We observe that the final ranking is the same as on ImageNet. Moreover, each model's improvement relative to prior models on ImageNot is roughly the same as on ImageNet. Figure 1 summarizes these results. The remarkable consistency of relative improvements suggests an intriguing counterfactual. *Had ImageNot, or other large-scale datasets with different properties, been the benchmark at the time of a new image classification model development, it could have induced the same improvements as ImageNet.* Crucially, we train from scratch on ImageNot and evaluate on an ImageNot test set, so we avoid the confound of model training that is pervasive in previous work to truly test external validity.

A major hurdle in executing this comparison is that architectures developed for ImageNet do not readily train well on ImageNot. Adapting each architecture to ImageNot requires some amount of hyperparameter tuning. To level the playing field between different architectures, we set careful a priori guardrails around hyperparameter tuning. Specifically, we only tune a small set of hyperparameters that is the same for all models. Moreover, all models get an equal tuning and training budget. These guardrails avoid confirmation bias and limit the researcher's degrees of freedom.

**ImageNot pretraining and transfer learning.** ImageNet has proved useful for pretraining and transfer learning (Kornblith et al., 2019). We wanted to know if the same is true for ImageNot. Toward this end, we conducted two experiments for each model:

1. Pretrain on ImageNot/ImageNet, finetune all layers on CIFAR-10, evaluate on CIFAR-10.

2. Pretrain on ImageNot/ImageNet, finetune last layer on CIFAR-10, evaluate on CIFAR-10 – *transfer*.

Similar to ImageNet, we find that a model's test accuracy on ImageNot is also a good predictor of fine-tunability and transferability.

To summarize, rankings of image classification models are surprisingly stable across significantly different datasets. Prior work had observed that small variations in a data-generating process preserve model rankings. Our work adds a more extreme contrast. Even significant deliberate deviations from a dataset can maintain essential characteristics of model training and comparison. In particular, our findings suggest that the external validity of machine learning is, in important ways, significantly greater than previously assumed. Code: `https://github.com/olawalesalaudeen/imagenot`.

## 2 Discussion of related work

**Early work on dataset bias.** The validity of benchmark results for image classification has long been of concern. Indeed, benchmarks often contain biases that do not generalize to data observed in the real world. Ponce et al. (2006) provide evidence of the narrow range of variation within existing benchmarks in the early stages of image classification research. Subsequently, Torralba and Efros (2011) compare multiple well-known image classification datasets, including SUN09 (Xiao et al., 2010), PASCAL Visual Object Classes (VOC) (Everingham et al., 2010), and Caltech-101 (Fei-Fei et al., 2004). They find evidence of selection bias, capture bias, and negative set bias in these datasets. Hardt and Recht (2022) Chapter 8 and (Liberman, 2015) give background and history of datasets and benchmarks.

**The ImageNet era.** The desire for a more representative benchmark for real-world image classification tasks motivated the creation of ImageNet, a large-scale image database (Deng et al., 2009). The extensive scale and careful curation of ImageNet aimed to alleviate some of the limitations identified in earlier datasets. ImageNet was at the center of over a decade of active development in image classification, largely centered around improving accuracy on the ImageNet ILSVRC 2012 competition dataset. The observed progress on image classification can be well summarized by a set of models that span the active development of model architectures around ImageNet and a couple of modern models developed after the challenged ended in 2017: **AlexNet** (Krizhevsky et al., 2012), **VGG** (Simonyan and Zisserman, 2014), **Inception V3** (Szegedy et al., 2015), **ResNet** (He et al., 2016), **DenseNet** (Huang et al., 2017), **MobileNet V3 Large** (Howard et al., 2019), **EfficientNet** (Tan and Le, 2019), **ViT-B-16** (Dosovitskiy et al., 2020), and **ConvNeXt** (Liu et al., 2022). While other models were also developed during the so-called ImageNet era, we focus on models that were most impactful and represented a notable shift in architecture design—we include a discussion in Appendix A. Specifically, we considered the availability of model implementations across hubs such as TorchHub (PyTorch Team, 2024), TensorFlow (Abadi et al., 2015), and Huggingface (Wolf et al., 2020) to be an indicator of popularity and, therefore, impact.

**Investigations about ImageNet.** Not least due to its excessive use over the years, researchers registered concerns about the external validity of ImageNet results. Tsipras et al. (2020) studied the design choices of ImageNet and their effect on alignment between the ImageNet dataset and the broader underlying object

recognition task. They concluded that ImageNet contains some inherent biases that cause misalignment. Despite this misalignment, they argue, improvements in multiclass classification on ImageNet still translate to the underlying object recognition task. However, they raised the issue that these biases make it difficult to identify actual improvements in object recognition. Xiao et al. (2020) identify one of these biases in that models tend to rely on the background of ImageNet images, which are correlated with the corresponding labels. Recht et al. (2019) investigated whether ImageNet classifiers generalize to a newly curated ImageNet test set—recall that by ImageNet, we generally refer to the ILSVRC 2012 ImageNet challenge. They found that accuracy decreases strongly from the original to the new test set, in the range of 11%-14%. Barbu et al. (2019) introduce a test set (ObjectNet) with real-world viewpoint, background, and rotation variation to assess model robustness and show that standard ImageNet-trained models suffer a 40–45% drop in top-1 accuracy on ObjectNet. Still, Recht et al. (2019) found that model rankings did not significantly change between the two ImageNet test sets: if *model a* performed better than *model b* on the original ImageNet test set, it generally also performed better on the new ImageNet test set. This speaks to the stability of model rankings under a slight distribution shift (new versus old test set), a form of *internal validity*. In other words, model rankings replicate under similar testing conditions. This form of internal validity is also supported by studies involving other variants of ImageNet. Hendrycks and Dietterich (2019) introduced ImageNet-C/P, ImageNet with common corruptions. Hendrycks et al. (2020) introduced ImageNet-R, stylized distribution shifts on ImageNet. Li et al. (2022) introduced ImageNet-W, ImageNet with watermarks, a learning shortcut. Shirali and Hardt (2023) introduced LAIONet, an intended recreation of ImageNet from the LAION-400M dataset, in order to identify key differences between ImageNet and LAIONet.

**External validity of ImageNet model rankings.** From prior work, we know that model rankings satisfy **internal validity**, i.e., they replicate in similar conditions, whereas accuracy numbers do not. In this work, we ask a daring question about the *external validity* of model rankings: Do model rankings replicate in radically different conditions? Rather than studying variants of ImageNet and other smaller and similar datasets to ImageNet, we purposefully create a deliberately different dataset, in a sense, an *anti-replication* of ImageNet. Surprisingly, we find that models from the ImageNet era achieve the same relative improvements on ImageNot over prior models as they did on ImageNet.

In contrast, we train models from scratch on the new dataset rather than reevaluate ImageNet-trained models on new test sets. One exception, perhaps closest to our study, is the important work of Kornblith et al. (2019), who examined how well the performance on ImageNet predicts performance on other vision datasets. They also train from scratch and find that on smaller and older datasets, such as DTD, VOC2007, or Caltech-101, model rankings do not replicate cleanly; however, they also find a strong correlation between ImageNet accuracy and accuracy on these datasets. While Kornblith et al. (2019) considered standard computer vision datasets at a time, we purposefully create a dataset as different as possible from ImageNet from a data source (LAION) not available then. The key takeaway from Kornblith et al. (2019) could have been that ImageNet is special insofar as models developed on ImageNet transfer well. Our main conclusion, in contrast, is that from the perspective of model benchmarking, there may be nothing all that special about ImageNet other than its scale and diversity. In particular, from the benchmarking perspective, our study suggests that we do not even seem to need clean labels (Beyer et al., 2020). This surprising takeaway was confirmed in a theoretical study (Dorner and Hardt, 2024).

## 3   Constructing ImageNot

We develop ImageNot, a dataset of the same scale as ImageNet's ILSVRC-2012—1000 classes with 1000 examples per class—but without key design choices that have been thought critical in ImageNet's effectiveness (Russakovsky et al., 2015). First, ImageNot has deliberately arbitrary classes constrained only to be strictly different from those in ImageNet. Second, we do not utilize any human annotators. There were no humans in the loop in selecting specific image and label pairs. We create the ImageNot dataset from the LAION-5B dataset, an academic, public massive scale collection of 5 billion image-text pairs crawled from the web and filtered by the CLIP model (Schuhmann et al., 2022; Radford et al., 2021). This contrasts the ILSVRC 2012 challenge, a subset of the ImageNet database which was constructed from Flickr search results (Deng et al., 2009).

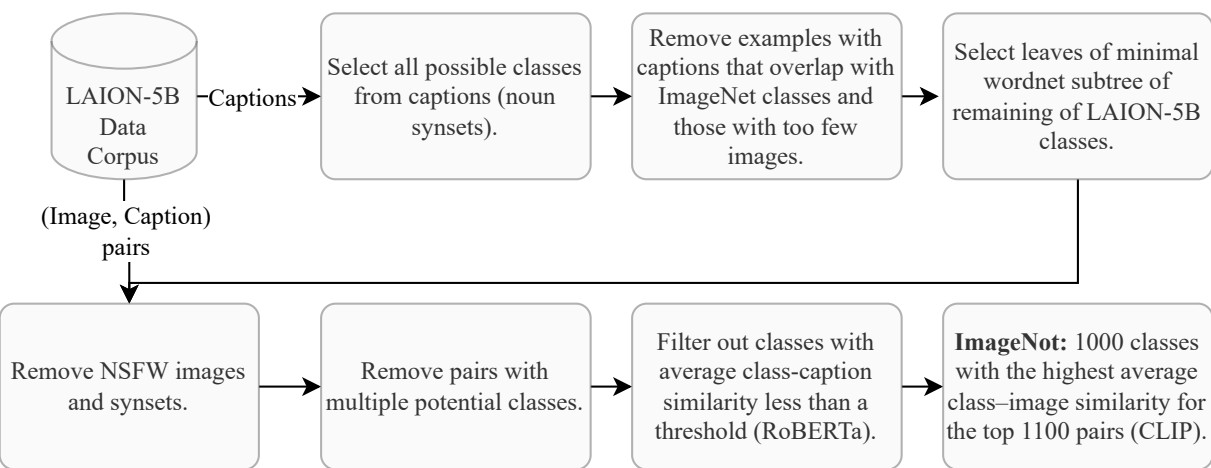

Figure 3: Overview of the ImageNot dataset curation pipeline. Starting from the LAION-5B corpus, the process selects all candidate noun synsets, removes classes overlapping with ImageNet or containing too few samples, filters out NSFW or ambiguous data, and ranks remaining classes by semantic (RoBERTa) and visual (CLIP) similarity. The final stage yields 1,000 ImageNot classes matched in scale but disjoint in concept from ImageNet, enabling evaluation of external validity.

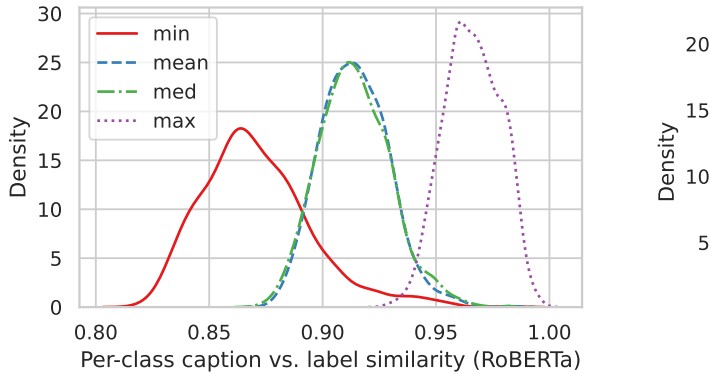

Figure 4: ImageNot caption x synset-gloss RoBERTa embeddings similarity for each class. We show some summary statistics of the similarity distribution across the 1000 classes.

Figure 5: ImageNot caption x synset-gloss RoBERTa embeddings similarity distribution for each class. Each distribution corresponds to similarities for each class.

ImageNot classes correspond to nouns in WordNet synsets found in LAION-5B captions, where WordNet is a large lexical database of English (Fellbaum, 1998; Miller, 1995; Hardeniya et al., 2016). The following definitions are important for describing WordNet. A given word has a set of *synsets*, where a synset is a sense of a word and has a corresponding short definition called a *gloss*. For example, 'car' has 2 synsets with corresponding glosses: (i) "a motor vehicle with four wheels; usually propelled by an internal combustion engine" and (ii) "a wheeled vehicle adapted to the rails of a railroad." Additionally, each synset consists of a set of synonyms called *lemmas*. The lemmas of 'car'-(i) include *automobile, motorcar*, while the lemmas of 'car'-(ii) include *railcar, railwaycar*. A WordNet tree represents a semantic hierarchy of synsets via an 'IS-A' structure, i.e., a synset is a 'type' of its synset parent, e.g., *mammal → placental → carnivore → dog → working dog → husky*.

At a high level, for each class, we select images from LAION-5B whose captions are similar to the synset corresponding to the class. We utilize RoBERTa (Liu et al., 2019) embeddings, an optimized version of BERT (Devlin et al., 2019), to estimate text sequence similarities. Since RoBERTa was never trained on

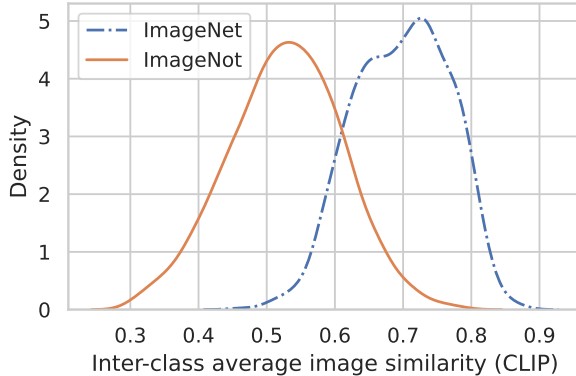

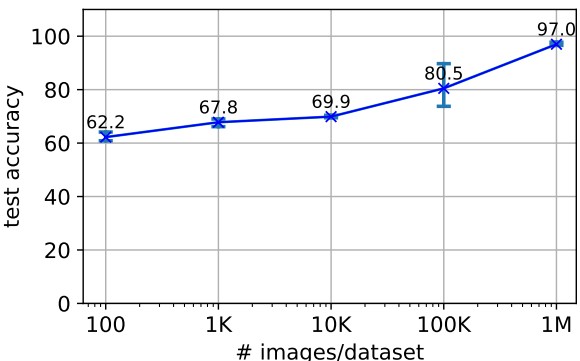

Figure 6: Image Variance. Average inter-class image similarity – inner product between normalized CLIP embeddings.

Figure 7: ConvNext distinguishes ImageNot vs. ImageNet images with high accuracy. Error bars are over 3 trials.

ImageNet, this selection mechanism mitigates the concern that our image selection process may be biased in favor of ImageNet-like images.

We note that creators of LAION-5B's only included image and caption pairs that are sufficiently similar based on CLIP embeddings (Contrastive Language–Image Pre-training (Radford et al., 2021)), a multimodal deep learning model (Schuhmann et al., 2022).

**Constructing ImageNot.** In creating ImageNot, we build on the data-generating process developed by Shirali and Hardt (2023). To choose the 1000 synsets (classes) that are included in ImageNot, available in the provided code, we take the following steps:

1. Find all *noun* synsets in the LAION-5B captions.

2. Remove all synsets that belong to any ImageNet synset/lemmas.

3. Filter synsets by size, dropping any candidate synset with too few image-caption pairs.

4. Find the minimal subtree of the WordNet synset tree that contains all of the remaining candidate synsets and select only the synsets that are leaves of this subtree.

5. Filter out synsets considered 'Not Safe for Work' (NSFW) by Song et al. (2021)'s NSFW classifier.

6. Find the subset of captions such that no caption contains multiple ImageNot synset's lemmas

7. Filter out synsets with an average top $n$ caption x synset-gloss pair RoBERTa embeddings similarity (normalized inner product) less than 0.82 (Liu et al., 2019). We found 0.82 to be the highest reasonable choice for sufficient examples in enough classes (Figure 4).

8. Finally, select the top 1000 classes ordered by the average similarity between the top 1100 class x image pairs (Radford et al., 2021).

Finally, we obtain ImageNot, a large-scale dataset that contrasts ImageNet. ImageNot has an average caption-label definition similarity of 0.91. Next, we compare ImageNot to ImageNet.

### 3.1 ImageNot vs. ImageNet

In this section, we examine the differences between ImageNot and ImageNet, focusing on their design aspects. The primary distinctions are: (i) the classes in ImageNot are deliberately far and different from those in ImageNet, and (ii) the attribution of a synset to an image in ImageNot is conducted without human intervention. We begin by examining the key differences between the two datasets.

**The 'Not' in ImageNot.** The first significant *not* is the *hierarchy* of synsets, based on the WordNet tree. ImageNet is constructed with synsets representing a densely populated subset of the WordNet tree. For instance, ImageNet contains 147 categories of dogs. In contrast, ImageNot is not designed to have densely populated subsets of the WordNet tree. Moreover, ImageNot synsets tend to be higher up in the WordNet tree (Appendix B.2 Figure 11) with ImageNot synsets having a median depth of 8 while ImageNet synsets have a median depth of 10 (more details about the WordNet subtrees can be found in Appendix B Table 3). One consequence of this difference in median depth is that the average diversity of images in an ImageNot synset is higher than that of ImageNet. We illustrate the distribution of intra-class similarity within each dataset in Figure 10, where ImageNot exhibits higher intra-class diversity than ImageNet.

Importantly, this difference does not mean that ImageNot and ImageNet occupy entirely disconnected regions of WordNet. Since WordNet is hierarchical, disjoint synsets can still share higher-level parent nodes. This parent overlap is not a class overlap. ImageNet and ImageNot have no shared training or evaluation classes by construction. Rather, the overlap reflects the coarse semantic organization of WordNet, while the datasets differ in which synsets they select and how densely those synsets populate the tree.

The second significant *not* is the *accuracy* of image-synset pairs. In ImageNet, an average precision of 99.7% is achieved across 80 synsets randomly sampled at different depths of the WordNet tree (Deng et al., 2009). Candidate images were collected from the internet by querying image search engines, and then human annotators were employed to verify each candidate image for a given synset, yielding relatively accurate image-synset pairs. In contrast, we deliberately make no such cleaning steps in ImageNot. We rely entirely on similarities between embeddings from CLIP (image and caption similarities) and RoBERTa (caption and class definition similarities) to select class and image pairs. Consequently, ImageNot's labels are far noisier than ImageNet, which utilizes human annotators. We estimate that 36% of images in ImageNot are either mislabeled (including degenerate images). Many mislabelings arise from images that match the colloquial meaning of the class word but not the WordNet synset definition. Degenerate cases include blank, stock, or filler images from image repositories; more details on the analysis and results can be found in Appendix B.1. We note that the 'noise' encompasses both incorrect labels due to automated curation and the greater natural variability of the images in ImageNot. This noisiness is noticeable in the accuracy observed when training on ImageNot – the best model achieves around 60%, which is drastically better than chance (0.1%) but lower than what the best model achieves on ImageNet (around 85%). We also see the effect of this process in the distribution of inter-class image similarities between ImageNot and ImageNet (Figure 6 and Appendix B.2 Figure 10); the similarity in images for an ImageNet class tends to be higher than those in an ImageNot class.

Finally, we find that a classifier distinguishes ImageNet and ImageNot well—a common strategy to examine the extent to which two datasets differ (Torralba and Efros, 2011; Liu and He, 2024). We use a ConvNext (Liu et al., 2022), randomly initialized, as a binary classifier of the two datasets and achieve a 97.0% accuracy when trained on the full training sets. Even with only 1000 samples, accuracy is significantly above chance, and classification accuracy increases with more samples, as shown in Figure 7. Trials include new data resampling and model initialization. We sample training data uniformly for each class except for N/class=1000. The test set is sampled similarly; however, the test images are sampled from the ImageNot/ImageNet test sets.

**Similarities in ImageNot and ImageNet WordNet subtrees.** Despite the differences in ImageNot/ImageNet, the subset of the WordNet trees spanned by their synsets is structurally similar. They share approximately the same number of nodes and edges, have similar depth, density and modularity, and share a 23% overlap in the parents of the datasets' synsets – illustrated in Appendix B.2 Figure 12 (more on WordNet subtrees can be found in Appendix B.2 Table 3).

## 4 ImageNot preserves model rankings

Our experimental design is centered around evaluating whether ImageNot maintains the model rankings established by ImageNet. Specifically, we want to know: *if a new architectural advancement had been evaluated on ImageNot instead of ImageNet, would it still have been recognized as a significant development?* Our focus is on a collection of model architectures that represent significant milestones in the evolution of ImageNet models. This includes **AlexNet** (Krizhevsky et al., 2012), **VGG** (Simonyan and Zisserman,

2014), **Inception V3** (Szegedy et al., 2015), **ResNet** (He et al., 2016), **DenseNet** (Huang et al., 2017), **MobileNet** (Howard et al., 2019), **EfficientNet** (Tan and Le, 2019), **Vision Transformers** (Dosovitskiy et al., 2020), and **ConvNeXt** (Liu et al., 2022)—see section 2 for motivating this set. For each model, we select the configuration that achieved the highest reported performance on ImageNet, such as ResNet152 over ResNet18. These models are then trained and evaluated on ImageNot, allowing us to directly compare their rankings with those obtained on ImageNet.

To test each model architecture, we train a version on ImageNet and another on ImageNot and evaluate them on the corresponding test set. We sort the test accuracy to establish a definitive order of performance in object recognition tasks, which we refer to as the 'model ranking.'

We also investigate ImageNet and ImageNot model performance for finetuning and transfer to the CIFAR10 dataset, which consists of 60,000 32x32 colored images of 10 classes, with 5000 training images per class and 1000 test images per class (Krizhevsky et al.; 2009). The learning task is 10-class image classification – Airplane, Automobile, Bird, Cat, Deer, Dog, Frog, Horse, Ship, and Truck. We note that ImageNet contains 50 test images per class (Deng et al., 2009) while ImageNot contains 100.

We consider three types of experiments: (i) *Random Initialization (From Scratch)*. We initialize each model with random weights, then train each model on a dataset, and finally, we evaluate each model on the same dataset's test set. (ii) *Fine-Tuning*. We initialize each model with pre-trained on ImageNot/ImageNet weights, then we fine-tune the model end-to-end (all weights updated) on CIFAR10, and finally, we evaluate it on CIFAR10's test set. (iii) *Transfer Learning*. We initialize each model with pre-trained ImageNot/ImageNet weights, and then we update only the model's last fully connected layer while training on CIFAR10. Finally, we evaluate it on CIFAR10's test set.

### 4.1 Model development

**ImageNet.** Our reference rankings are derived from pre-trained models available on *Torch Hub* (Marcel and Rodriguez, 2010; Paszke et al., 2019). These models establish a ranking, which we consider our *factual* observations. In decreasing order of performance, the ranking is as follows: EfficientNet V2 Large, ConvNeXt Large, Vision Transformer B-16, ResNet 152, Inception V3, DenseNet 161, MobileNet V3 Large, VGG 19, and AlexNet—Figure 1. Please refer to Appendix A.1 for additional specifics regarding these pre-trained models. We also utilize the implementation and pre-trained model weights from torch hub (Marcel and Rodriguez, 2010; Paszke et al., 2019) for fine-tuning and transfer with ImageNet pretraining.

**On avoiding self-fulfilling prophecies.** If not careful, one may fall into a subtle pitfall – the inadvertent confounding effect of knowing the desired ImageNet ranking. Specifically, an experimenter aware of the expected ranking might unconsciously adjust the process until a particular model is in the right place. This iterative tuning could falsely lead to a preservation of the known rankings. To mitigate this risk, we implemented a priori restrictions on the experimenter's approach for each ranking modality (see Appendix A.1): (1) there is a fixed number of hyperparameter runs that can be launched per model (budget) and (2) the hyper-parameter search space for all models is expanded only if some model(s) do not train. For ImageNot, we start with the hyperparameters used to train reference ImageNet models to maintain objectivity. However, this means that our resulting training configuration for each model may not be optimal (Figure 8a-8b) relative to the decade-plus of tuning to obtain the best ImageNet models.

### 4.2 Model Rankings

**ImageNot preserves model rankings and relative improvements.** Figure 1 illustrates that the model rankings between ImageNot and ImageNet are remarkably consistent. We compare each model's performance improvement relative to AlexNet, the model that kick-started the deep learning era on ImageNet in 2012. This comparison reveals a notable parallel in the progression of accuracy improvements across both ImageNet and ImageNot. Not only do the ordinal rankings of models remain unchanged, but the degree of performance enhancement is also consistent. In simpler terms, this suggests that *if a new architecture like EfficientNet (Tan and Le, 2019) had been introduced in a counterfactual universe focused on ImageNot instead of ImageNet, we would have witnessed a comparable impressive advancement.* The external validity of ImageNet improvements

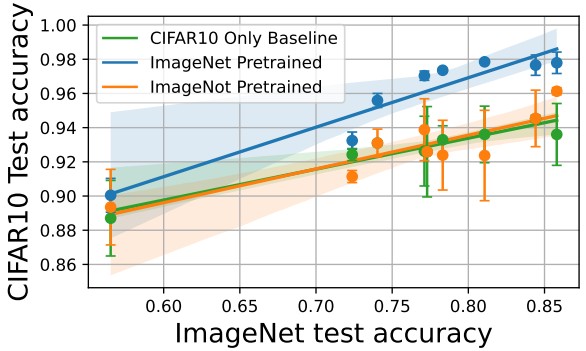 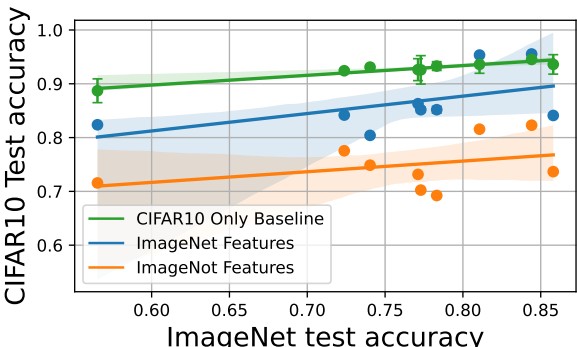

(a) **Fine-tuned** to CIFAR10—test accuracy compared to standard ImageNet test accuracy for each model architecture

(b) **Transfer Learning** to CIFAR10—test accuracy compared to standard ImageNet test accuracy for each model architecture.

Figure 8: ImageNet vs. ImageNot. Fine-tuning and Transfer Learning with 95% confidence for regression estimates. Robust corresponds to average accuracy across test-time distribution shifts—error bars do not show for sufficiently small intervals.

follows. The ImageNet vs. ImageNot relative test accuracies have a linear fit with a slope of 1.01 and a Pearson R of 0.99. We note that these slopes should not be interpreted the same way as those in Recht et al. (2019) because we compare test accuracies of models with different training tasks/data with different difficulties.

Our results also highlight the limitations of relying solely on the external validity of accuracy as the primary metric for the effectiveness of a benchmark. ImageNet model accuracies range from 0.57 (AlexNet) to 0.85 (EfficientNet V2 L), while much noisier ImageNot (Figure 2) model accuracies approximately range from 0.4 (AlexNet) to 0.6 (EfficientNet V2 L). One may look at the drop in accuracy from ImageNet to ImageNot and conclude that ImageNet results have no external validity, but this would be a rather narrow and inaccurate conclusion. From the standpoint of benchmarking, the more relevant observation and notion of external validity is that relative progression in ImageNet accuracy is nearly mirrored by progression in ImageNot accuracy, and model rankings are replicated exactly, despite the significant difference between the two datasets.

For CIFAR10, we do three random restarts and compute the standard errors for each model's test accuracy. We find that rankings are also preserved for CIFAR10 when training from scratch, up to error bars (Figure 9). Kornblith et al. (2019) also evaluate CIFAR10 and find a similarly strong correlation between ImageNet accuracy and CIFAR10 accuracy. Given that we opted to include architectures that achieve very similar accuracies on the ImageNet test (such as EfficientNet-V2-L/ConvNeXt-L or ResNet152/DenseNet161), we expect potential training variance to affect comparisons, particularly given the size of CIFAR10. For instance, when training CIFAR10 from randomly initialized weights or ImageNot (noisy), we observe overlapping error bars for a majority of the best models across random restarts. However, the consistency of relative rankings still holds on average.

Our methodology restricts experimenters such that once a hyperparameter search space and the number of samples from the search space are fixed, they cannot be adjusted unless a model does not train. This approach safeguards the rankings from confirmation bias. However, we note that it also has some notable limitations, e.g., models with similar capabilities with overlapping error bars.

**ImageNot improvements predict improvements in fine-tuning and transfer learning.** When fine-tuning pre-trained ImageNot/ImageNet models on CIFAR10, we observe that improvements in ImageNot predict improvements in finetunability of ImageNot models—similar to ImageNet and ImageNet models (Figure 8a). The ImageNot test accuracy and test accuracy of ImageNot models fine-tuned to CIFAR10 have a slope of 0.26 and a Pearson R of 0.83. Finetuned ImageNet has a slope and Pearson R of 0.29 and 0.95,

respectively. Here, we find that, as a benchmark (our scope), ImageNot is counterfactually near-equivalent to ImageNet, but as a pretraining dataset, ImageNet is superior.

When transferring ImageNot vs. ImageNet features, we find that ImageNet features are notably more performant (Figure 8b). This suggests that the noisiness of ImageNot does have some negative downstream effects on transfer learning. However, we still observe a positive trend in ImageNot test accuracy and transferability of ImageNot models to CIFAR10—slopes of 0.30 and 0.32 with ImageNot and ImageNet, respectively.

**Findings hold under robustness.**   We find that relative improvement also remains closely aligned between ImageNet and ImageNot when we look at average performance across 17 random data augmentations applied to the ImageNot/ImageNet test set (Figure 9; Appendix A.3), e.g., random blurring and cropping. ConvNeXt outperforms EfficientNet V2, however, this holds for both ImageNet and ImageNot, and MobileNet V3 performs marginally better than expected on ImageNot under these augmentations. Nevertheless, our results support the claim that preserving absolute metrics across datasets can be a misleading notion of external validity.

**Safety and ethical limitations of web-crawled datasets.**   While our results suggest that web-crawled datasets with minimal human intervention may be sufficient for benchmarking progress in model development, such datasets raise serious ethical and safety concerns. As illustrated by Thiel (2023), the internet contains illegal images and images of illegal activity. In particular, Thiel (2023) previously found evidence of Child Sexual Abuse Material (CSAM) in LAION-5B. We guard against this risk by using an NSFW language classifier (Song et al., 2021) to remove synsets that may be associated with harmful images, in addition to relying on the default LAION-5B NSFW filter. These safeguards are necessarily imperfect. We did not manually inspect every image in ImageNot, and model-based filtering cannot guarantee the absence of harmful, sensitive, or mislabeled content. For this reason, ImageNot should not be understood as a curated collection of safe images or as a general-purpose dataset for unrestricted use. We provide ImageNot class names, a detailed description of our generation procedure, and code so that our results can be reproduced without presenting ImageNot as a fully curated image archive. Downstream users who reconstruct the dataset should apply their own safety checks before using, inspecting, or redistributing any images.

## 5   Conclusion

Despite the drastic differences in ImageNet and ImageNot, we find consistent preservation of model rankings across both datasets. Moreover, this consistency extends to pre-trained models' fine-tunability and the transferability of pre-trained image embeddings. For benchmarking, we argue that providing robust model rankings, rather than absolute accuracy numbers, should be the primary function of a good benchmark. Consequently, our results suggest that concerns about the external validity of model developments on ImageNet as a benchmark may be overstated.

Many domain experts attribute the rapid advancement in object recognition models to the ImageNet benchmark. From this work, two takeaways emerge to guide and motivate future benchmarks. First, a benchmark that produces robust model rankings may be effective even if accuracy degrades under distribution shifts. Second, clean labels may not be necessary to measure progress effectively.

## Acknowledgements

We are indebted to Ali Shirali for providing his code for generating datasets from LAION, which we modified to create ImageNot. We thank André Cruz, Ricardo Dominguez-Olmedo, Florian Dorner, and Katherine Tsai for many helpful discussions on this project and invaluable feedback on an earlier version of this paper. We also thank Sanmi Koyejo for helpful discussions on this project. Finally, we would like to thank the Action Editor and the anonymous reviewers at TMLR for their constructive feedback and insightful comments, which helped improve the clarity and quality of this paper.

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

# A  Architectures and Training

**Architecture Development Trajectory.**  **AlexNet** (Krizhevsky et al., 2012) demonstrated the efficacy of deep learning and many architectural advances such as convolutional filters, rectified linear units (ReLUs) as activation functions, and more. Additionally, its relative success on the ILSVRC 2012 is considered to have revitalized interest in neural networks, sparking a wave of research in deep learning. **VGG** (Simonyan and Zisserman, 2014) then implemented a simpler and deeper architecture with smaller convolutional filters, along with regularization techniques such as dropout, which yielded improved accuracy and generalization. **Inception V3** Szegedy et al. (2015) focused on multi-scale feature extraction by learning convolutions of different sizes in parallel. Additionally, auxiliary classifiers were added at intermediate layers to enhance gradient flow and act as implicit regularizers. **ResNet** (He et al., 2016) further improved upon AlexNet and VGG with the introduction of residual blocks, which allowed for the training of deeper networks while mitigating the vanishing gradient problem and achieving superior performance to its predecessor. **DenseNet** (Huang et al., 2017) proposed to connect each layer to every other layer to maximize information flow between layers. **MobileNet V3 Large** focused on optimizing deep learning for mobile and embedded devices using depth-wise separable convolutions and squeeze-and-excitation (SE) modules (Hu et al., 2018). **EfficientNet** (Tan and Le, 2019) then improved upon its predecessors with compound and systematic scaling of depth, width, and resolution, as well as combinations of novel optimization and learning strategies, without increasing computational cost. **Vision Transformers (ViT-B-16)** apply a fundamentally different approach, replacing convolutional feature extractors with self-attention mechanisms; images are treated as sequences of patches, but self-attention across patches allows for long-range dependencies and flexible learning of spatial relationships. **ConvNeXt** (Liu et al., 2022) aimed to compete with the state-of-the-art performance of Vision Transformers (ViTs; (Dosovitskiy et al., 2020)) by combining components of preceding ConvNet architectures, e.g., residual blocks and kernel sizes.

## A.1  ImageNot Models

For ImageNot, we utilize hyperparameters reported for the corresponding ImageNet model to match the model configuration. More details of the hyperparameters used are specified in our Code, with reference to https://github.com/pytorch/vision/tree/main/references/classification. We only change these parameters if models do not train.

**AlexNet.**  The AlexNet model uses a step scheduler and the SGD optimizer. The learning rate is set to 0.01 with a minimum learning rate of 0.0. The learning rate decay rate is 0.1, with no warmup epochs and an auxiliary learning rate decay after 30 epochs. The model is trained for 90 epochs with a momentum of 0.9 and a weight decay of 0.0001. The batch size is 32.

**ConvNeXt Large.**  The ConvNeXt Large model uses a cosine annealing with linear warmup scheduler and the AdamW optimizer. The learning rate is 0.001 with a minimum learning rate of 0.0. The learning rate decay rate is 0.1, and warmup is applied for 5 epochs. The model is trained for 600 epochs with a momentum of 0.9 and a weight decay of 0.05. The batch size is 128, and label smoothing is set to 0.1. Additionally, mixup is set to 0.2, cutmix to 1.0, repeated augmentation sampling (RA sampler) is enabled with 4 repetitions, and exponential moving average (EMA) decay is set to 0.99998 with 32 EMA steps.

**DenseNet-161.**  The DenseNet-161 model uses a step scheduler and the SGD optimizer. The learning rate is 0.1 with a minimum learning rate of 0.0. The learning rate decay rate is 0.01, with no warmup epochs and an auxiliary learning rate decay after 30 epochs. The model is trained for 90 epochs with a momentum of 0.9 and a weight decay of 0.0001. The batch size is 32.

**EfficientNet V2-L.**  The EfficientNet V2-L model uses a cosine annealing with linear warmup scheduler and the SGD optimizer. The learning rate is 0.5 with a minimum learning rate of 0.0. The learning rate decay rate is 0.1, and warmup is applied for 5 epochs. The model is trained for 600 epochs with a momentum of 0.9 and a weight decay of 0.00002. The batch size is 32, and label smoothing is set to 0.1. Additionally,

mixup is set to 0.2, cutmix to 1.0, RA sampler is enabled with 4 repetitions, and EMA decay is set to 0.99998 with 32 EMA steps.

**Inception V3.** The Inception V3 model uses a step scheduler and the SGD optimizer. The learning rate is 0.1 with a minimum learning rate of 0.0. The learning rate decay rate is 0.1, with no warmup epochs and an auxiliary learning rate decay after 30 epochs. The model is trained for 90 epochs with a momentum of 0.9 and a weight decay of 0.0001. The batch size is 32. Additionally, the auxiliary loss weight is set to 0.4.

**MobileNet V3-Large.** The MobileNet V3-Large model uses a step scheduler and the RMSProp optimizer. The learning rate is 0.064 with a minimum learning rate of 0.0. The learning rate decay rate is 0.973, with no warmup epochs and an auxiliary learning rate decay after 2 epochs. The model is trained for 600 epochs with a momentum of 0.9 and a weight decay of 0.00001. The batch size is 128.

**ResNet-152.** The ResNet-152 model uses a step scheduler and the SGD optimizer. The learning rate is 0.1 with a minimum learning rate of 0.0. The learning rate decay rate is 0.1, with no warmup epochs and an auxiliary learning rate decay after 30 epochs. The model is trained for 90 epochs with a momentum of 0.9 and a weight decay of 0.0001. The batch size is 32.

**ViT-B/16.** The ViT-B/16 model uses a cosine annealing with a linear warmup scheduler and the AdamW optimizer. The learning rate is 0.001 with a minimum learning rate of 0.0. The learning rate decay rate is 0.033, and warmup is applied for 30 epochs. The model is trained for 300 epochs with a momentum of 0.9 and a weight decay of 0.3. The batch size is 512, and label smoothing is set to 0.11. Additionally, gradient clipping is set to 1.0, mixup is set to 0.2, cutmix to 1.0, RA sampler is enabled with 3 repetitions, and EMA decay is set to 0.99998 with 32 EMA steps.

**VGG-19.** The VGG-19 model uses a step scheduler and the SGD optimizer. The learning rate is 0.01 with a minimum learning rate of 0.0. The learning rate decay rate is 0.1, with no warmup epochs and an auxiliary learning rate decay after 30 epochs. The model is trained for 90 epochs with a momentum of 0.9 and a weight decay of 0.0001. The batch size is 32.

We utilize early stopping with patience of 10 based on held-out validation sample accuracy and set a maximum epoch of 1000 for all training. Additionally, to optimize training efficiency and reduce computational costs, we employ the Asynchronous Hyper-Band Scheduler from *Ray Tune* (Li et al., 2020; Liaw et al., 2018), which enables us to further early-stop less promising hyperparameters.

**Compute Resource Estimate.** We give a table of approximate times to reproduce our results below.

Table 1: Compute Time. Note that this only accounts for the time it took to train the models used to produce the results in this paper.

| Dataset | Results | GPU-hrs | GPUs |
|---|---|---|---|
| ImageNot | From Scratch | 800 | 8 A100 |
| CIFAR10 | From Scratch | 40 | 2 A100 |
| CIFAR10/ImageNet | Finetune | 8 | 2 A100 |
| CIFAR10/ImageNot | Finetune | 12 | 2 A100 |
| CIFAR10/ImageNet | Transfer Learning | 4 | 2 A100 |
| CIFAR10/ImageNot | Transfer Learning | 6 | 2 A100 |

## A.2 CIFAR10 Models

For both fine-tuning and transfer learning from ImageN[o/e]t, we use these pretrained weights as our starting point and then update weights either by fine-tuning or transfer learning. We also train end-to-end only on CIFAR10 with randomly initialized model weights. The hyperparameter details can be found in Table 2.

Table 2: Hyperparameters search space and Training Budgets for experiment. TL: Transfer Learning to CIFAR10; FT: Finetuning to CIFAR10; ST: Standard, randomly initialized weights trained only on CIFAR10.

| Dataset | Hyperparameter | Train Style | Range | Budget |
|---------|----------------|-------------|-------|--------|
| CIFAR10 | batch size | TL/FT/ST | [2048/256/256]* | 24 |
| CIFAR10 | optimizer | TL/FT/ST | [SGD, Adam] | 24 |
| CIFAR10 | lr | TL/FT/ST | [1e-3, 1e-1] | 24 |
| CIFAR10 | lr scheduler | TL/FT/ST | [Cosine Annealing with Warm Restarts] | 24 |
| CIFAR10 | momentum | TL/FT/ST | [0.8, 0.99] | 24 |
| CIFAR10 | optimer warmup steps | TL/FT/ST | [5, 10, 15, 25, 50] | 24 |
| CIFAR10 | weight decay | TL/FT/ST | [1e-6, 5e-3] | 24 |

### A.3 Robustness

We employ 17 distinct test-time data augmentations to evaluate the robustness of the developed models, all random except for the Identity augmentation. These augmentations are applied to the same set of held-out test examples. They include: (1) *Identity*, which uses the original image; (2) *Gaussian Blur*, applying a Gaussian blur to smooth the image and reduce noise; (3) *Adjust Sharpness*, for modifying image sharpness; (4) *Affine*, performing affine transformations such as scaling, rotations, and translations; (5) *ColorJitter*, adjusting brightness, contrast, saturation, and hue; (6) *Contrast*, altering contrast levels; (7) *Equalize*, for histogram adjustment and contrast balancing; (8) *Grayscale*, converting the image to grayscale; (9) *Horizontal Flip*, flipping the image horizontally; (10) *Invert*, inverting pixel values; (11) *Perspective*, applying perspective transformations; (12) *Posterize*, reducing the bit depth per color channel; (13) *Resized Crop*, cropping after resizing; (14) *Rotation*, rotating the image; (15) *Sharpness*, for sharpness adjustment; (16) *Solarize*, inverting colors post a threshold; and (17) *Vertical Flip*, flipping the image vertically.

## B  Additional Results

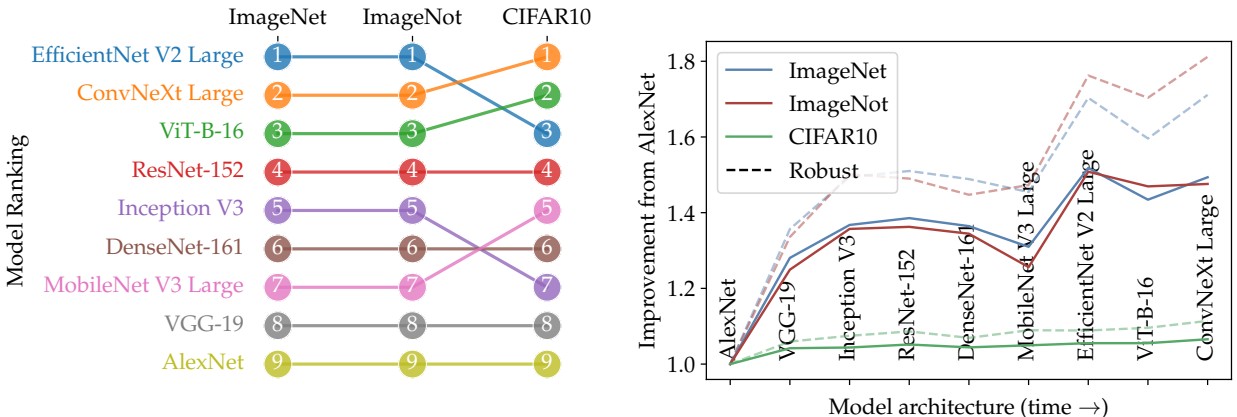

Figure 9: Final Model Ranking and Relative Improvement. Model rankings and relative improvements hold with models trained from scratch on the respective dataset, and evaluated on a held-out test set. For a set of test accuracies $X$, relative progress is $X_i/X_{\text{AlexNet}}$. ImageNet model accuracies range from 0.57 (AlexNet) to 0.85 (EfficientNet V2 L), while much noisier ImageNot model accuracies approximately range from 0.4 (AlexNet) to 0.6 (EfficientNet V2 L). CIFAR10 accuracies range from 0.89 (AlexNet) to 0.95 (ConvNeXt Large). The rankings for CIFAR10 are derived from means across three restarts; we note that ranking differences from ImageNet only occur for models with error bounds overlapping with the model in its ImageNet position. Thus, even for CIFAR10, the rankings are preserved up to error. Robust corresponds to average accuracy across many test-time distribution shifts.

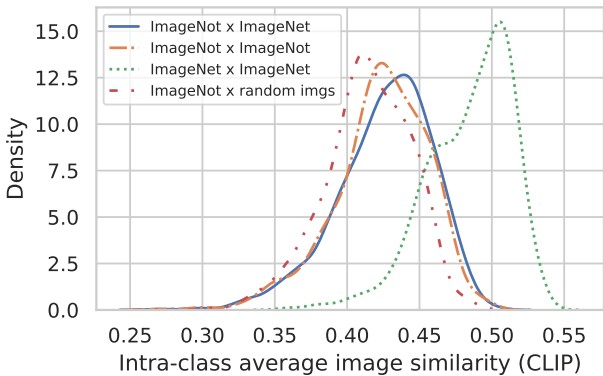

Figure 10: Class Diversity. Average intra-class image similarity – inner product between normalized CLIP embeddings.

### B.1 ImageNot Label Noise

The original ImageNet paper reports high annotation quality, with an average precision of 99.7% across 80 synsets randomly sampled at different depths of the WordNet tree (Deng et al., 2009). Subsequent work has nevertheless shown that ImageNet is not free of label defects; some examples have incorrect or ambiguous labels, and correcting these labels can noticeably change measured model performance (Beyer et al., 2020; Northcutt et al., 2021; Stock and Cisse, 2018). ImageNot is far noisier than ImageNet; an estimated 36% of images are mislabeled. Despite this large difference in label quality, ImageNot preserves the same model rankings and relative improvements observed on ImageNet. For the benchmarking question studied here, clean labels do not appear necessary to support the relative model development trajectory induced by ImageNet.

We audited the quality of ImageNot labels on a sample of 20 WordNet synsets, with 100 images per synset, for a total of approximately 2,000 images. Each image was rated independently by three vision-language models: Claude Sonnet 4.5, GPT-4, and Gemini 2.5 Flash-Lite. All three models used the same prompt, and we took the majority vote as the consensus annotation. The raters agreed substantially with one another, with Fleiss' $\kappa = 0.84$ for label correctness. We define correctness with respect to the assigned WordNet gloss, rather than the ordinary meaning of the class word. Under this strict synset-level criterion, 64% of images are correctly labeled by the consensus, i.e., a 36% label noise rate.

To check this automated audit, a human expert annotated the same images while blind to the model votes. The expert agreed with the consensus annotation on 97% of images, with Cohen's $\kappa = 0.71$. The remaining disagreements mostly reflect a simple ambiguity in ImageNot; the class word and the assigned WordNet sense need not coincide. For example, an image of a nuclear power plant matches the ordinary meaning of *reactor*, but not the assigned WordNet gloss, "an electrical device that introduces reactance." Thus, some images that look correct at the word level are incorrect under the stricter synset-level criterion; human annotators were less strict in this dimension. Even with 36% of the images mislabeled, under this strict definition, ImageNot preserves the same model rankings and relative improvements observed on ImageNet.

### B.2 ImageNet vs. ImageNot WordNet subtree

Table 3 summarizes the differences (or lack thereof) between properties of the ImageNet and ImageNot WordNet synsets (referenced in section 3.1). Figures 11 and 12 further depict the difference in ImageNot and ImageNet's WordNet subtree structure.

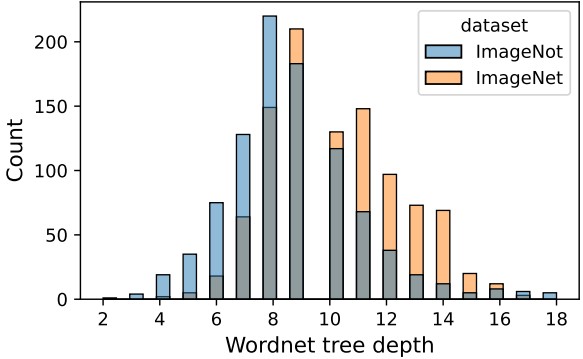

Figure 11: Synset depth in WordNet tree.

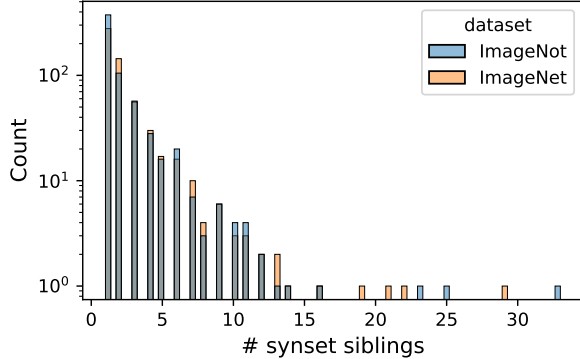

Figure 12: Number of synset siblings.

Table 3: ImageNot and ImageNet (ILSVRC 2012) WordNet tree statistics.

| Stat | ImageNot | ImageNet | $\frac{\text{ImageNot}}{\text{ImageNet}}$ |
|---|---|---|---|
| Nodes | 1877 | 1860 | 1.01 |
| Edges | 1899 | 1937 | 0.98 |
| Min Depth | 2 | 4 | 0.50 |
| Median Depth | 8 | 10 | 0.80 |
| Max Depth | 18 | 17 | 1.06 |
| Density | 5.39e-4 | 5.60e-4 | 0.96 |
| Modularity | 0.94 | 0.93 | 1.01 |
| Synset Overlap | 0% | 0% | – |
| Parent Overlap | 23% | 23% | – |

