# OpenReview forum: "ImageNot: A contrast with ImageNet preserves model rankings"
_TMLR — Accepted by TMLR_

### Review · Reviewer_nCgQ · 2026-02-13

**Summary Of Contributions:**

The authors test the external validity of the benchmarking process in image classification. They collect a new dataset similar to ImageNet, with the same number of classes, training images, and image domain (i.e., general images), employing a different data-generation procedure. Then, the authors retrain several impactful vision architectures (using corresponding training procedures) on this dataset. The results reveal that the model rankings (and, moreover, improvement rates) do not change across datasets, suggesting that model development benchmarks are, in some sense, externally robust to the data-generating process.

**Strenghts**
1. The research question seems important for the machine learning community.
2. The methodology seems reasonable; the results are easily interpretable.

**Weaknesses**
1. The scope of the study is rather narrow. The authors focus on assessing external validity using highly similar datasets.
2. The findings are not surprising given the previous literature on the accuracy-on-the-line phenomenon and related topics.

**Audience:**

Yes

**Audience Explanation:**

As I outlined in the summary, the question seems important for the community. In particular, this topic is important for designing benchmarking procedures.

**Broader Impact Concerns:**

I do not feel the need to add a Broader Impact Statement. However, the paragraph on "Safety and ethical limitations of web-crawled datasets" might be highlighted more if the authors plan to release the dataset.

**Claims And Evidence:**

Yes

**Claims Explanation:**

The evidence seems sufficient for the scope of the experiments. However, I think that the authors should interpret their results more carefully.

1. Even though the authors provide additional CIFAR-10 experiments, the main focus of the paper is performance comparison of architectures on ImageNet and ImageNot. As I outlined in the summary, these datasets are highly similar. The authors test external validity only with respect to class similarity and label noise. This comparison omits such important parameters as the dataset size, the number of classes, the domain, and the learning task itself (e.g., it is still unclear whether relative rankings would hold if the dataset was smaller, had fewer classes, and was focused on a specific domain, such as the medical domain).
2. While the authors claim the stability of model rankings, this interpretation is not entirely correct. In Appendix A.1, we see that the training procedures differ across architectures. Thus, in addition to architectural advancements, the results also reflect improvements in "model training", making interpretation more nuanced.
3. While the results show that milestone improvements hold across datasets, it is less clear whether smaller improvements will also be properly reflected. This question seems important since a lot of hyperparameter tweaking leads to modest gains (e.g., see PyTorch blog [1]). Thus, it remains unclear whether the precise structure of ImageNet enabled these tweaks to emerge.

[1] https://pytorch.org/blog/how-to-train-state-of-the-art-models-using-torchvision-latest-primitives/

**Requested Changes:**

1. Please clarify the scope of external validity that you considered in the paper.
2. Please discuss whether you expect your findings to hold for less significant tweaks.
3. Please clarify the meaning of "architecture design".

---

### Review · Reviewer_CizJ · 2026-02-19

**Summary Of Contributions:**

The paper asks a simple question: are the ImageNet-era model rankings we all know actually specific to ImageNet, or would the same relative ordering show up on another large-scale classification dataset that is deliberately different from ImageNet?

To test this, the authors built a new dataset called ImageNot. It matches ImageNet in scale (1000 classes, 1000 images per class) but tries hard to avoid overlap with ImageNet categories and to rely on a different data/labeling pipeline (web data with automated filtering rather than the original ImageNet curation). They then train a set of representative architectures from scratch on ImageNot and compare their test accuracies and rankings to the same models trained/evaluated on ImageNet. The headline result is that the rankings are essentially unchanged, and the “relative improvement” curves across architectures closely track each other between ImageNet and ImageNot (near-linear agreement). The paper also includes transfer experiments (pretrain on ImageNot vs ImageNet, then fine-tune/linear-probe on a downstream dataset) to see how far the conclusions extend beyond in-domain classification.

Key strengths: The question is clearly stated and the experiment is conceptually clean: construct a large dataset that is intentionally “not ImageNet,” then ask whether rankings persist when training from scratch. The dataset scale match makes the comparison easy to interpret, and the paper adds guardrails to reduce experimenter degrees of freedom in hyperparameter tuning. The main finding is straightforward and well supported by the reported correlation/near-linear relationship in performance.

Key weaknesses: In my view the main contribution is largely retrospective. The field already has broad evidence, across many downstream tasks and benchmarks, that modern architectures and training recipes generalize better than older ones; so showing that ImageNet rankings transfer to one more large web-derived classification dataset feels incremental and somewhat dated. Also, ImageNot is derived from LAION-style pipelines and uses automated filtering, which raises questions about dataset bias, label noise, and how representative it is of “non-ImageNet” vision problems in general. Finally, because the core claim is about rankings, the paper would benefit from more emphasis on statistical uncertainty (seeds/variance and whether close models are actually distinguishable) and clearer guidance on ethical/data-governance issues when releasing or using such web-scraped datasets.

**Audience:**

Yes

**Audience Explanation:**

Yes. The paper touches a topic that many in the TMLR community care about: how much we should trust benchmark-driven conclusions, and whether ImageNet rankings reflect something real beyond a single dataset. The ImageNot construction and the finding that rankings stay similar can still be useful for researchers working on dataset shift, benchmark validity, and evaluation methodology, even if the overall contribution is more retrospective than forward-looking.

**Broader Impact Concerns:**

My main broader impact concern is dataset safety and governance. ImageNot is constructed from large-scale web-crawled data (LAION-style), and such sources can contain harmful or illegal content (for example, sexual content, exploitation material, or sensitive personal data). Even if the paper’s goal is benchmarking rather than deployment, releasing a new dataset (or enabling easy reconstruction via URLs/IDs) can create pathways for misuse and can expose annotators and users to harm. The submission should more clearly state what will be released, what filtering and safety checks were applied (and their limits), and what precautions are recommended for downstream users.

**Claims And Evidence:**

No

**Claims Explanation:**

First, ImageNot is built from LAION-style web data with an automated filtering pipeline, and parts of the construction rely on model-based similarity signals. This makes it hard to rule out selection effects that could favor the same kinds of representations and training recipes that already work well on ImageNet. In other words, the dataset may be “different in labels,” but still similar in the features that modern models exploit, so ranking stability is not necessarily strong evidence of external validity.

Second, the dataset appears to contain substantial label noise and ambiguous class-image matching by design. When the paper’s conclusion is about fine-grained ranking, the analysis should more clearly quantify uncertainty: run-to-run variance, confidence intervals, and whether close model pairs are statistically distinguishable. Without that, “the same ranking” can be an artifact of noise, limited seeds, or hyperparameter choices rather than a robust property.

Finally, the paper largely focuses on large-scale image classification. Given that architecture progress has already been validated across many downstream tasks, it is not clear that this specific re-test provides strong new evidence about general progress, beyond one additional web-derived classification benchmark. Overall, the results are suggestive, but I do not think they yet meet the bar of clear and convincing evidence for the broader claim being made.

**Requested Changes:**

1. Report results over multiple random seeds for each model on ImageNot and ImageNet, and provide confidence intervals / standard errors.

2. The current ImageNot pipeline uses automated similarity/filtering steps on LAION-style data. Please add controlled ablations showing how rankings change if you remove or weaken key steps (for example: no CLIP-based filtering/reranking, different text-model for caption–gloss matching, different similarity thresholds). The goal is to demonstrate that the ranking stability is not an artifact of the specific filtering recipe.

3. Provide a more direct measurement of label quality: sample a subset of classes/images for human verification, estimate noise rate, and analyze whether noise varies by class.

4. Revise the framing to clearly state what is and is not supported: the evidence is about large-scale image classification on one web-derived dataset with noisy labels, not about “external validity” in general or across tasks (detection/segmentation/long-tail/open-vocabulary).

---

### Review · Reviewer_hPny · 2026-03-05

**Summary Of Contributions:**

The paper introduces ImageNot, an image classification dataset sampled from LAION-5B that differ from ImageNet while approximately matching its scale (1K classes, 1K images per class). Using this dataset, the authors investigate whether the historical progress of model architectures observed on ImageNet would hold under a different benchmark. They train several representative architectures on ImageNot and compare their rankings to those on ImageNet. The results show that model rankings and relative improvements are largely preserved across the two datasets.

**Additional Comments:**

N/A

**Audience:**

No

**Audience Explanation:**

**Findings:**

Model rankings remaining stable across datasets has been reported many times before, even for distribution shifts [1,2,3], and even for LAION dataset comparisons [4]. I understand that the main contribution of this paper is training models from scratch on a new dataset but the "new" dataset is roughly coming from the same overall distribution (see my comment above on the dataset). So, I fail to see how this work is of any particular interest to readers who are familiar with the literature.

[1] Recht et al. Do ImageNet Classifiers Generalize to ImageNet

[2] Kornblith et al. Do Better ImageNet Models Transfer Better

[3] Taori et al. Measuring Robustness to Natural Distribution Shifts in Image Classification

[4] Shirali et al. What Makes ImageNet Look Unlike LAION

**Claims And Evidence:**

No

**Claims Explanation:**

**Issues with the dataset creation:**

Although ImageNot is presented as substantially different from ImageNet, the two datasets remain similar in several key aspects. The label space in both cases is derived from WordNet synsets, preserving a similar semantic structure. Furthermore, both datasets consist of natural images collected from the web that contain everydy objects and scenes. As a result, despite differences in specific classes and labeling procedures, ImageNot still represents the same natural image classification regime as ImageNet, and therefore does not constitute a fundamentally different test of external validity. Even based on authors admission, there's about 20% overlap between ImageNet and ImageNot despite all the work put in to make them dissimilar. Had this dataset instead consisted of images from substantially different domains such as medical or biomedical imaging (e.g., radiology, histopathology), satellite or aerial imagery, microscopy, industrial inspection images, autonomous driving sensor data, underwater imagery, astronomical observations, or scientific imaging modalities, the conclusions about the robustness of model rankings might have been more compelling, as such settings would introduce genuinely different visual distributions and task characteristics.

**Issues with the evaluation:**

The authors claim to evaluate performance differences between model architectures, but in practice the experiments evaluate not only the models themselves but also the training recipes and hyperparameters used to train them. Many of the more recent architectures are typically trained with substantially improved optimization strategies, data augmentation schemes, regularization techniques, and longer training schedules compared to earlier models. Of course the newer models will perform better with newer training routines and newer augmentations. A clear example of this phenomenon is demonstrated in the *ResNet Strikes Back* paper [1], where the authors show that when older architectures such as ResNet are trained using modern training recipes, their performance improves significantly and can approach or match that of more recent models.

Here is an example:

> AlexNet is trained with SGD, a step learning rate schedule, and a relatively short 90-epoch training schedule with minimal regularization.

> In contrast, ConvNeXt Large uses a much more modern training recipe including AdamW optimization, cosine learning-rate scheduling with warmup, label smoothing, MixUp, CutMix, repeated augmentation sampling, and a much longer 600-epoch training schedule.

Was there any doubt that their ranking wouldn't change between two datasets?

**Tone of the paper:**

The tone of the paper is somewhat stronger than warranted by the scope of the experiments. Several statements frame the results in terms of the external validity of machine learning progress or counterfactual developments in model architecture, even though the empirical evaluation is limited to a single alternative dataset and a small set of architectures within the same natural image classification domain.

The authors also frame the experiments as large-scale, but this is somewhat misleading by today's standards. Except for the top three models, most architectures evaluated in this work were introduced nearly a decade ago, and training on ImageNet-scale datasets is relatively modest compared to many studies that train much larger models on substantially larger datasets.

**Other Minor Issues:**

There are too many emphasis (\emph{}) and the tone of the paper is quite bombastic, very nontraditional, which I believe hurts readability.

There are several strong statements in the paper that are presented without citations and appear overstated. For example, the opening claim that "models developed in one domain consistently perform worse in new domains" is not well supported and is somewhat misleading. While performance degradation under distribution shift is well documented, prior work has shown that models with higher ImageNet accuracy generally transfer better to other datasets and tasks [3]. In practice, improvements on ImageNet have often translated into improvements across a variety of downstream vision benchmarks, even when absolute accuracy drops under domain shift. A more nuanced statement, supported by appropriate citations, would better reflect the existing literature.

Dataset labels and tree structure could be clarified further since there is still space in the paper.

Misclassifications of models could be further discussed. Literature is clear that in many classes, ImageNet failures stem from mislabeling issues [2] and fixing those improves the models performance noticeably.


[1] Wightman et al. ResNet strikes back: An improved training procedure in timm

[2] Beyer at al. Are we done with ImageNet?

[3] Kornblith et al. Do Better ImageNet Models Transfer Better?

**Requested Changes:**

I don't think the findings of this work are of much interest to researchers familiar with the developments of the past decade. As mentioned above, it could be more interesting to evaluate models on an ImageNet-scale dataset drawn from a substantially different domain rather than another collection of natural images, though this would admittedly require a significant effort.

In its current form, the work remains fairly high-level and does not provide deeper analysis or insights that would meaningfully advance our understanding of model behavior or benchmark design. As a result, it is unclear to me what new knowledge this study adds beyond observations that have already been documented in prior work.

---

### Review · Reviewer_UfBp · 2026-03-06

**Summary Of Contributions:**

The authors propose a novel benchmark to test whether the results obtained on ImageNet over the past decade are specific to it, or are inherently tied to the architectures tested. The benchmark is designed to overlap as little as posible with ImageNet, thus providing a dataset that, at least in theory, has little to no confounds with that dataset.

**Additional Comments:**

I believe the authors should for example take a look at research such as [1] which instead of using datasets of naturalistic images, used stimuli designed from classical experiments in Psychology. These are in many ways fundamentally different from ImageNet and show results that challenge the effectiveness of these models. However, they lack the scale to be useful training data for these models so they are mainly used with fine-tuning or inference.


[1] Biscione, V., Yin, D., Malhotra, G., Dujmovic, M., Montero, M. L., Puebla, G., ... & Bowers, J. S. (2024). MindSet: Vision. A toolbox for testing DNNs on key psychological experiments. arXiv preprint arXiv:2404.05290.

**Audience:**

No

**Audience Explanation:**

I don't think so. We already have enough experimental evidence to know which architectures are good at what.

**Claims And Evidence:**

No

**Claims Explanation:**

I think the main claim as phrased in the second paragraph (i.e. "How different a dataset from ImageNet would have supported essentially the same model developments over the years?") is not really supported by the experiments. The authors just construct ImageNot from a different set of 1000 classes which doesn't change the nature of the dataset in a qualitatively interesting way: it is still an object recognition dataset on naturalistic images and thus likely shares the same visual features as ImageNet. In other words I am not sure this eliminates the confounds that are important to answer the question.

**Requested Changes:**

I am not sure there are any changes that can make this work pass review.

---

### Decision · Action_Editor_MZho · 2026-04-21

**Recommendation:** Accept with minor revision

**Additional Comments:**

This submission creates a new 1000-class image dataset, which the authors call ImageNot, from the LAION-5B dataset. Its main contribution is to show that the relative accuracy of architectures/training recipes is preserved between ImageNet and ImageNot.

Reviewers were somewhat split on the value of the submission, noting that its scope is somewhat limited. Although I agree with this assessment, I am not aware of published work that has done exactly what this submission has done. While the fact that the correlation between ImageNet and ImageNot accuracy is high is not in itself surprising, it is nonetheless useful to do the comparison and see how big the effect is, and to have more data points supporting the generalization of benchmark results on ImageNet to other natural image datasets. Reviewers also had some concerns regarding the framing of the results, but I believe that these concerns can be addressed in a minor revision.

Overall, I support acceptance. The authors should make the changes that they promised the reviewers, specifically:
- Caveating usage of the term "external validity"
- Clarifying that the recipes compared encompass more than architecture
- Including the described human study of label quality
- Extending the discussion of dataset safety, limitations, and the WordNet label structure

**Audience:**

Yes

**Audience Explanation:**

A large body of literature has shown that model accuracy on ImageNet is highly predictive of accuracy for transfer learning and accuracy on test images under distribution shift. However, I am unaware of any work that has thoroughly explored the correlation with ImageNet accuracy when *both training and evaluating* on an independently collected ImageNet-like dataset. Although the results here are not especially surprising and thus the significance of the work seems modest, I believe it meets TMLR's criteria.

**Claims And Evidence:**

Yes

**Claims Explanation:**

The submission's main claim, that different models perform similarly in relative terms on both ImageNot and ImageNet, is convincingly demonstrated. Three reviewers raised concerns that the type of "external validity" that the submission assesses was not defined and could be misinterpreted. Two reviewers noted the submission uses the word "architecture" to refer to what is really a combination of an architecture and a training recipe, which is inaccurate, but that inaccuracy is within the scope of what can be corrected in a minor revision.